# Soccer-related injuries utilization of U.S. emergency departments for concussions, intracranial injuries, and other-injuries in a national representative probability sample: Nationwide Emergency Department Sample, 2010 to 2013

Gerardo Flores [1]*, Christopher C. Giza[2‡], Barbara Bates-Jensen[1,3‡], Mary-Lynn Brecht[1‡], Dorothy Wiley[1]

1 School of Nursing, University of California Los Angeles, Los Angeles, CA, United States of America,
2 Departments of Neurosurgery and Pediatrics, University of California Los Angeles, Los Angeles, CA, United States of America, 3 David Geffen School of Medicine, University of California, Los Angeles, CA, United States of America

☯ These authors contributed equally to this work.
‡ These authors also contributed equally to this work.
* gerardo.flores@ucla.edu

## Abstract

Soccer participation in the United States (U.S.) has increased over time, and injuries as well as interest to prevent injuries has become more common. This study described Emergency Department (ED) visits related to concussions, intracranial injuries (ICI), and all-other injuries attributed to soccer play; described healthcare cost and length of hospital stay of soccer-related injuries; and determined independent predictors of concussions, ICI, and all-other soccer injuries leading to ED visits. The study examined soccer-related weighted discharge data from the Nationwide Emergency Department Sample, Healthcare Cost and Utilization Project, Agency for Healthcare Research and Quality. Weighted tabular analysis of univariate and bivariate analyses and weighted and adjusted logistic regression models were conducted. A total of 480,580 of U.S. ED visits related to soccer injuries were available for analysis between 2010 to 2013. Generally, 98% of soccer-related ED visits resulted in routine (treat-and-release) visits. However, the odds of transfer to a short-term hospital following ED evaluation and treatment was more than 37-fold higher for soccer-injured youth and adults diagnosed with ICI when compared to all-other soccer injuries; additionally, these patients showed 28-fold higher odds of being admitted for inpatient care at the ED-affiliated hospital. For concussion, soccer-injured patients with concussion showed nearly 1.5-fold higher odds of being transferred to a short-term hospital than did those with any other soccer injury. Soccer-related ED visits cost more than 700 million in U.S. dollars from 2010 to 2013. Notable differences were noted between concussions, ICI, and all-other soccer injuries presenting to U.S. ED. Albeit underestimated given that this study excludes other forms of health care and treatment for injuries, such as outpatient clinics, over the

**Data Availability Statement:** The data underlying the results presented in the study are available from the Healthcare Cost and Utilization Project (HCUP) - Nationwide Emergency Department Sample (https://www.hcup-us.ahrq.gov/).

**Funding:** The author(s) received no specific funding for this work.

**Competing interests:** The authors have declared that no competing interests exist.

counter medications and treatment, and rehabilitation, healthcare cost associated with soccer-related injuries presenting to ED is high, and remarkably costly in those with an ICI diagnosis.

## Introduction

Soccer is the most popular sport in the world, with an estimated 265 million soccer players worldwide, nearly 4% of the worlds' 7.4 billion population [1, 2]. The United States (U.S.) ranks second worldwide, in recreational, non-recreational, and professional soccer participation, with continued increase in participation from 10.3 million in 1993, 13 million in 1999, to over 24 million current adult and youth players [1, 3, 4]. Registered youth participation in organized soccer leagues in the US increased 88% in just over 25 years, with over 3 million U.S. youth participating in soccer in 2014 vs 1.6 million in 1990 [1, 5]. With increased U.S. soccer participation since the early 1990's, soccer-related injuries have become more common and are of interest to minimize human suffering, disability, healthcare costs, and utilization.

The spectrum of soccer-related injuries varies widely from sprains and strains to traumatic brain injuries (TBI), with great and different concerns seen between concussions and other intracranial brain injuries (ICI), due to the potential for long-term effects and chronic disability. Data suggest that participation in sports contributes to an estimated 1.6 to 3.8 million TBI's annually, with most sports-related TBI's diagnosed as mild, 85–86% [6–8]. The prevalence of mild-TBI are conservative estimates as many go unreported [6–8]. The lasting adverse effects of TBI's can result in poor school health [9–12]. Children that suffered a mild-TBI exhibit a 9-fold higher rate of disability, when compared to moderate- and severe-TBI, and receive federally mandated programs for disability, such as special education, learning assistance, learning and attention issues accommodations, as well as tutoring, occupational, physical, and speech therapy [13]. In addition, when compared to controls, children with TBI received more mandated services after 12-months [13]. For that reason, there is an interest in describing and comparing soccer-related concussion, ICI, and all-other soccer injuries.

The Nationwide Emergency Department Sample (NEDS) datasets includes national probability sample representative of soccer-related injuries leading to U.S. emergency department (ED) visits. This national representative sample allows analysis of individual-level (e.g. age, gender, insurance coverage, household income), timing (e.g. day of the week, month of the year), hospital-level (e.g. teaching status, trauma designation), geographic region and population density characteristics, mechanism and place of injury, and healthcare utilization characteristics of soccer-injuries.

The primary purpose of this study was to describe soccer-related concussions, ICI, and all-other soccer injuries. A secondary purpose was to describe healthcare utilization, cost and length of hospital stay of soccer-related injuries. A third purpose was to determine independent predictors (e.g. age, gender, and other individual and well as hospital-level) of concussions, ICI, and all-other soccer injuries leading to ED visits.

## Methods

Exemption from the University of California, Los Angeles Institutional Review Board was obtained to conduct a secondary analysis from a pre-existing dataset (2017; IRB#17–000777). The study was classified as *not human research* by the UCLA South Campus Institutional Review Board. Specifically, the data has anonymous observations, devoid of any identifiers

that could link these results to institutions or persons. In addition, we have abided by HCUP requirements to maintain confidentiality.

## Study design and sample

The study examined emergency department visits for soccer-related visits using discharge data from the NEDS, Healthcare Cost and Utilization Project (HCUP), Agency for Healthcare Research and Quality [14]. The 2010 to 2013 NEDS datasets provide serial cross-sectional characteristics of adults and children treated in a stratified sample of rural, suburban and urban U.S. hospital-affiliated ED's. Of these, we evaluated data for all soccer-related injuries, including concussions and ICI treated at ED's. NEDS has been described extensively elsewhere [14]. Briefly, NEDS provides national estimate of all-payer, population-level data for U.S. ED visits for four U.S. regions (north, south, east, west), urban and rural locations, and care setting characteristics including teaching and non-teaching hospitals; private, public or government ownership; and level of trauma care provided in the affiliated ED's. NEDS sites are selected from stratified and clustered eligible U.S. ED's, and a 20% weighted random sample of treated individuals are selected for record abstraction. Weighted sampling allows generalizability to the U.S. population. The NEDS dataset includes 100% of records selected. Records are not linked at the individual level over time or across institutions; consequently, individuals may contribute more than one study visit annually or over many years, and it does not allow tracking of individuals over time. Over the study period, illnesses and injuries were coded using International Classification of Disease, 9th Revision, Clinical Modification (ICD-9-CM) codes for each ED visit. NEDS provides 1 to 15 diagnosis (outcome) codes, ≤4 external cause of injury codes (E-codes), as well as hospital and patient characteristics. Subjects with an E007.5 ICD-9 E-code were selected for the analysis. Data for 480,580 weighted individual soccer-injury ED visits completed from 2010 thru 2013 were evaluated for associations between socio-demographic, hospitalization, and care-costs characteristics for concussive and ICI in comparison to all other soccer injuries.

## Variables

Subjects were classified into three groups, concussion, ICI, or all-other-soccer injuries based on the 15 ICD-9 provider coded diagnoses placed in the medical record. For all soccer-related injuries, the prevalence of ICI, with or without concussion, and concussion alone were compared to all-other soccer-related injuries. Only 6% (85/1369) of ICI-diagnosed patients were diagnosed with concussion and ICI; due to the scarce data, we included cases with concussion and ICI into the ICI group. Descriptors, disposition, diagnostic outcomes, mechanism of injury, place of injury, healthcare utilization, and independent predictors were analyzed between concussions, ICI, and all-other soccer injuries.

   Descriptors included year, gender, age, median household income, primary payer, month of the year of ED visit, day of the week of ED visit, hospital region, hospital teaching status, hospital trauma designation, hospital population density based on location, and whether (or not) presenting with multiple injuries for all soccer-related injuries in NEDS. NEDS includes age in years for cases presenting to the ED and gender as male or female. Age in years were categorized as: ≤ 6 years of age (y/o), 7–11 y/o, 12–17 y/o, 18–24 y/o, 25–34 y/o, 35–44 y/o, 45 + y/o. NEDS estimates patient median household income from the patient's residential zip codes and provides a variable that categories patients into four-quartiles from the poorest to the wealthiest populations, 0–25th, 26th–50th, 51st–75th, and 76th - 100th. The primary payers included (1) public health insurance, (2) private health insurance, (3) self-pay (e.g. uninsured), and (4) other (e.g. worker's compensation, other government programs such as CHAMPUS,

CHAMPVA, Title V). We also looked at the 12-months of the year as well as the day of the week (weekday versus weekend) to determine at what period in time soccer-related injuries are most likely to occur. The hospital characteristics included (1) *geographical region* (Northeast, Midwest, South, and West); (2) *teaching designation* was categorized into a dichotomous variable, where metropolitan non-teaching hospitals were combined with all non-metropolitan hospitals, to maintain confidentiality of patients due to the scarce number of non-metropolitan teaching hospitals (metropolitan teaching status versus metropolitan non-teaching hospitals or non-metropolitan); (3) *hospital trauma designation* was generally categorized into trauma vs not trauma, but we had to keep a third level on this variable (non-trauma and trauma level III) as NEDS purposefully collapsed any stratum that had less than two trauma hospitals to maintain hospital confidentiality; (4) *hospital population density* was divided into large-metropolitan ($\geq$ 1 million residents) versus other population density areas, such small metropolitan, micropolitan, and urban-rural, we took this approach as there was scarce data in some of these areas, since 90% of the ED visits occurred in metropolitan (small and large) areas. We also examined if presenting with multiple injuries (vs not) changed the effect of presenting to the ED with a soccer-related injury.

Other descriptors of interest included hospital disposition, such as routine visits, transfer to short-term hospital, transfer to other facility (includes skilled nursing facilities, intermediate care facilities, and other types of facilities), home health care, leave against medical advice, admitted as inpatient to the same hospital, and died in emergency department. Disposition refers to the destination of the patient upon discharge directly from the ED. This excludes the destination of patients following discharge from a hospitalization; inpatient admission. For example, a patient that was admitted as an inpatient to the same hospital as the ED, could later be discharged home or left against medical advice during hospitalization. We only examined disposition upon discharge from the ED, not inpatient stay. We were interested in understanding the destination of people with soccer-related injuries from ED.

ICD-9-CM codes were used to explore the effects of mental health conditions on concussions, ICI, and all-other injuries. There was special interest in Attention Deficit Disorders and other mental health conditions, categorized as binary variables (yes or no).

ICD-9-CM codes were also used to identify the location of injury, mechanism of injury, and place of injury, which were coded as binary variables (yes or no). We looked at head and neck, trunk, upper extremity, and lower extremity injury locations. Based on the available administrative data, mechanism of injury was determined using three binary variables, yes vs. no: 1) falls; 2) struck by hit or thrown/kicked ball, without subsequent fall; 3) struck by hit or thrown/kicked ball, with subsequent fall. Place of injury was determined using six binary variables: 1) Home, 2) recreational and sports facility, 3) street and highway, 4) public institution (e.g. school), 5) residential institution, and 6) other (e.g. beach).

Descriptors for healthcare utilization included cost and length of inpatient stay. NEDS includes two different variables for cost: total charges for ED services, and total charges for those hospitalized (ED and inpatient services). ED charges were analyzed for all soccer related injuries. For those hospitalized, combined ED and inpatient charges (e.g. inpatient charges), and hospital length of stay were analyzed. NEDS charges are in U.S. dollars and allowable charges range from $75 to $75,000 in 2010, and $100 to $950,000 from 2011 to 2013; any case outside of the allowable charges were excluded from analysis by design. The length of stay is calculated by subtracting the discharge date from the admission date and it is reported in days, from 0 to 365 days. Any same-day admissions are coded as 0. Also note, NEDS reports as invalid any inpatient admissions that exceed the maximum allowed length of stay; as a result, these cases are excluded from the analysis.

Sociodemographic, timing of the injury, geographic and health care resources, as well as multiple injuries (vs not) were used as independent predictors variables for adjusted logistic regression models.

## Analysis

NEDS provides a stratum and cluster variable; we included these variables on our analyses to account for the effects of sampling, and a weight variable to report weighted national estimates from probability sample. SAS statistical software was used to manage data, conduct statistical analyses of weighted frequencies, charges sum, means, as well as adjusted logistic regression analyses of independent predictors of injury for concussions, ICI, and all-other soccer-related injuries.

Tabular and descriptive analysis were used to report weighted frequencies and percentages for categorical variables: socio-demographics, geographic and health care resources, disposition, mental health conditions, injury location, mechanism of injury, and place of injury. Given the scarce data of some categories for disposition, inferential analyses were not conducted. For all other categorical variables, chi-square statistics were used to determine if there are significant differences between levels on outcome of interest (concussion, ICI, and all-other soccer injuries). Significance was set at $p \geq .05$.

Bivariate associations were explored in a series of logistic regression models where variables were progressively added in the order listed in Fig 1 (not reported). Associations between concussions and intracranial injuries, (each) compared to all other soccer-related injuries, were estimated using multivariable adjusted logistic regression analyses to calculate odds ratios and 95% confidence intervals (Fig 1). Thus, the adjusted model simultaneously adjusted for the effects of ED visit year, gender, age (category), median household income (quartiles), primary payer type, month, weekday and weekend days, and hospital characteristics, including region, trauma center designation, metropolitan (vs. other), and multiple (vs. single) injury characteristics (Fig 1). Next, we evaluated the associations between mental health conditions (2), injury locations (3), mechanisms of injuries (3) and places of injury (4) for concussions or intracranial injuries, each in comparison to all other soccer-related injuries. In these comparisons, the reference population includes all others that were not exposed to the condition, location, mechanism or place of injury evaluated. For each, we simultaneously adjusted for the effects of sociodemographic, hospital characteristics, and the presence of multiple (vs. single) injuries detailed in Fig 1. In all adjusted models, we included case weighting and proportional distributions based on NEDS probability sampling characteristics. Generally, the variable level with the greatest proportion of ED visits was set as the reference group, with the exception of year and month variables, where the reference groups were each set to 2010 and January, respectively.

Last, the sum of charges and means in U.S. dollars, with standard deviations, were calculated for ED charges and inpatient charges for concussions, ICI, and all-other injuries related to soccer. Tabular analysis on excel were conducted to determine percentages of total ED visits, ED charges, and total inpatient charges across diagnostic outcomes.

## Results

All soccer-related injuries represent 480,580 of U.S. Emergency Departments (ED) visits evaluated between 2010 to 2013, which account for less than 1% of all-injuries and all ED visits, and nearly 11% of all-sports injuries. Our analysis targeted all soccer-related injuries. Among soccer-related injuries presenting to U.S. ED's, the majority were male (304,175 [63.3%]), with 12–17 y/o representing the largest age group (234,133 [48.7%]). The largest proportion of

| Characteristics | Concussions | | | Intracranial injuries | | |
|---|---|---|---|---|---|---|
| | OR | 95% CI Low | High | OR | 95% CI Low | High |
| **Year** | | | | | | |
| 2010 | ref | | | ref | | |
| 2011 | 1.13 | 1.01 | 1.27 | 0.83 | 0.54 | 1.28 |
| 2012 | 1.28 | 1.15 | 1.43 | 0.60 | 0.39 | 0.93 |
| 2013 | 1.30 | 1.16 | 1.46 | 0.54 | 0.34 | 0.86 |
| **Gender** | | | | | | |
| Male | ref | | | ref | | |
| Female | 1.16 | 1.09 | 1.24 | 0.56 | 0.40 | 0.79 |
| **Age** | | | | | | |
| ≤ 6 years | 0.47 | 0.36 | 0.61 | 1.02 | 0.37 | 2.82 |
| 7-11 years | 0.50 | 0.46 | 0.55 | 0.54 | 0.33 | 0.87 |
| 12-17 years | ref | | | ref | | |
| 18-24 years | 0.67 | 0.61 | 0.73 | 1.06 | 0.73 | 1.53 |
| 25-34 years | 0.37 | 0.32 | 0.42 | 1.25 | 0.83 | 1.90 |
| 35-44 years | 0.27 | 0.22 | 0.33 | 1.04 | 0.62 | 1.75 |
| 45+ years | 0.25 | 0.18 | 0.33 | 1.27 | 0.64 | 2.50 |
| **Median Household Income** | | | | | | |
| 0-25th percentile | 0.69 | 0.61 | 0.78 | 0.82 | 0.53 | 1.28 |
| 26th-50th percentile | 0.70 | 0.63 | 0.78 | 1.10 | 0.76 | 1.57 |
| 51st-75th percentile | 0.82 | 0.75 | 0.88 | 0.78 | 0.56 | 1.08 |
| 76th-100th percentile | ref | | | ref | | |
| **Primary Payer** | | | | | | |
| Public | 0.52 | 0.47 | 0.57 | 0.58 | 0.38 | 0.89 |
| Private | ref | | | ref | | |
| Self-pay | 0.73 | 0.64 | 0.83 | 0.91 | 0.62 | 1.34 |
| Other | 0.87 | 0.75 | 1.01 | 1.49 | 0.86 | 2.59 |
| **Month of the year** | | | | | | |
| January | ref | | | ref | | |
| February | 0.89 | 0.76 | 1.06 | 0.93 | 0.43 | 1.99 |
| March | 0.89 | 0.75 | 1.04 | 1.48 | 0.81 | 2.71 |
| April | 0.93 | 0.79 | 1.08 | 0.79 | 0.39 | 1.60 |
| May | 0.91 | 0.78 | 1.05 | 0.97 | 0.48 | 1.93 |
| June | 0.71 | 0.59 | 0.85 | 1.31 | 0.68 | 2.49 |
| July | 0.64 | 0.53 | 0.78 | 1.03 | 0.50 | 2.12 |
| August | 0.85 | 0.72 | 1.00 | 0.94 | 0.49 | 1.79 |
| September | 1.02 | 0.88 | 1.17 | 0.89 | 0.48 | 1.66 |
| October | 1.15 | 1.00 | 1.32 | 0.89 | 0.48 | 1.66 |
| November | 1.04 | 0.89 | 1.22 | 0.99 | 0.52 | 1.87 |
| December | 1.13 | 0.94 | 1.36 | 0.46 | 0.18 | 1.21 |
| **Day of the week** | | | | | | |
| Weekday | ref | | | ref | | |
| Weekend | 0.95 | 0.90 | 1.01 | 1.14 | 0.90 | 1.46 |
| **Hospital region** | | | | | | |
| Northeast | 1.06 | 0.93 | 1.21 | 0.66 | 0.40 | 1.10 |
| Midwest | 1.11 | 0.99 | 1.26 | 0.92 | 0.60 | 1.41 |
| South | 1.25 | 1.09 | 1.43 | 1.33 | 0.81 | 2.18 |
| West | ref | | | ref | | |
| **Hospital Teaching status** | | | | | | |
| Metropolitan non-teaching | ref | | | ref | | |
| Metropolitan teaching | 1.00 | 0.89 | 1.11 | 1.15 | 0.80 | 1.63 |
| **Hospital Trauma Designation** | | | | | | |
| Not trauma | ref | | | ref | | |
| Trauma | 1.21 | 1.09 | 1.34 | 1.27 | 0.91 | 1.78 |
| Non-trauma and trauma | 1.01 | 0.87 | 1.18 | 1.19 | 0.72 | 1.97 |
| **Hospital Location** | | | | | | |
| Large Metro | ref | | | ref | | |
| Other | 1.14 | 1.03 | 1.26 | 0.78 | 0.53 | 1.13 |
| **Multi-injury** | | | | | | |
| Yes | 2.92 | 2.70 | 3.16 | 7.55 | 5.73 | 9.94 |
| No | ref | | | ref | | |

*Simultaneously adjusting for the effects of sociodemographic, geographic and health care resources. Weighted case estimates and proportional distributions based on probability sampling.

**Fig 1. Adjusted odds of concussion or intracranial injuries (each) versus all other soccer injuries in a population-based sample of persons receiving evaluation and care at U.S. emergency departments for soccer related injuries.**

patients belonged to the highest median household income (176,537 [37.3%]), presented with private medical insurance (293,418 [61.2%]), and in the beginning of the season, September (61,314 [14.5%]). With the general characteristics of all soccer injuries in mind, differences were noted when soccer-injuries were categorized into concussions, intracranial injuries (ICI), and all other-soccer injuries.

Period effects are notable among soccer-related concussions, ICI, and all other-soccer injuries. Generally, all-other soccer injuries accounted for 93.76% of ED visits and increased annually from 2% to 11%, with the largest increase, 11%, from 2011 to 2012 and smallest increase, 2%, from 2012 to 2013. Concussions totaled nearly 6% of all soccer injuries and increased annually from 5% to 24%. For example, the greatest increase in soccer-related concussion prevalence, 24%, was shown in 2011. The smallest increase, 5%, was shown in 2012. Conversely, ICI were rare in these data, 0.28%, and over this period, the prevalence of ICI decreased annually. Year to year, ICI prevalence decreases 1% to 24%. Overall, the notable period effects were taken into consideration for adjusted logistic regression, as well as gender and age.

Males were nearly three-fold more likely to be diagnosed ICI, and nearly two-fold more likely to present with any non-cranial injury than females. Albeit statistically significant, concussions among males (54%) and females (47%) were less disparate than gender specific ratios for ICI (76% vs. 24%) and other non-cranial injuries (64% vs. 36%). However, concussions make up a larger percentage of female injuries (7.5%) than males (5%). The highest proportion of injuries occurred in the 12–17 y/o for all other-soccer injuries, concussions, and ICI (Table 1). Overall, the odds of ED evaluation or treatment for concussion was highest for 12–17 y/o youth, compared to every other age group, ranging from 1.5- to 4-fold higher odds (Fig 1). Alternatively, 12–17 y/o youth evidenced higher odds for ICI only in comparison to 7–11 year old's (OR = 1.85 (1.15, 3.03) (Fig 1).

Generally, higher socio-economic status was associated with increased likelihood of attending the ED following a soccer-related injury, especially when diagnosed with a concussions or ICI. However, when diagnosed with ICI, the likelihood is not as prominent. Overall, patients reporting the highest income, 76th-100th percentile, or private insurance were treated in the ED for soccer-related injuries (Table 1). Patients diagnosed with concussions admitted to the ED were generally wealthier (46.0%) or were insured privately (74.9%), respectively; however, the wealthiest and privately insured soccer-related ED-evaluated patients represented only 36.7% and 60.36% of those treated for all other (soccer) injuries. Nearly 40% of soccer-related ICI diagnoses reported the highest income level; as well, 64.68% were privately insured.

Trends demonstrated that competitive play, such as the beginning of league and weekend games may increase the likelihood of soccer-related injury ED visits. During the month of September (beginning of soccer league season) all other-soccer injuries were nearly three-fold higher when compared to January when the soccer season is near the end, whereas concussions were over three-fold, and ICI over two-fold when comparing September and January. In some months, the likelihood was nearly six-fold, such as ICI in September compared to December. While not significant, the majority of injuries occurred Monday through Friday, during weekday play compared to play occurring on Saturday or Sunday, weekend days. However, play falling on weekend days resulted in a higher proportion of all soccer-related injuries for all-other, concussion-, and ICI-related injuries when compared to weekday play: 52%, 53%, and 84%, respectively. Although the month and day of the week may influence soccer-related ED visits, number of presenting injuries may also be contributors.

Concussions and ICI demonstrated notable differences in likelihood of presenting to U.S. ED's with multiple injuries following soccer play. For example, concussions were significantly nearly three-fold more likely to present with multiple injuries when compared to all other-

**Table 1. Associations between sociodemographic, geographic and health care resources for a population-based sample of patients presenting to U.S. emergency dpartments for evaluation and care for soccer related injuries (weighted case estimates and proportional distributions based on probability sampling).**

| Characteristics | All other injuries | | Concussions | | Intracranial injuries | | p-value |
|---|---|---|---|---|---|---|---|
| | n | % | n | % | n | % | |
| Number of soccer injuries | 450,587 | 93.76% | 28,624 | 5.96% | 1,369 | 0.28% | |
| Year | | | | | | | <0.0001 |
| 2010 | 102,290 | 22.70% | 5,447 | 19.03% | 414 | 30.20% | |
| 2011 | 107,588 | 23.88% | 6,548 | 22.88% | 379 | 27.67% | |
| 2012 | 119,090 | 26.43% | 8,130 | 28.40% | 290 | 21.17% | |
| 2013 | 121,619 | 26.99% | 8,499 | 29.69% | 287 | 20.96% | |
| Gender | | | | | | | <0.0001 |
| Male | 287,821 | 63.88% | 15,311 | 53.50% | 1,043 | 76.17% | |
| Female | 162,715 | 36.12% | 13,310 | 46.50% | 326 | 23.83% | |
| Age | | | | | | | <0.0001 |
| ≤ 6 y/o | 8,711 | 1.93% | 297 | 1.04% | 21 | 1.52% | |
| 7–11 y/o | 78,749 | 17.48% | 3,232 | 11.29% | 101 | 7.36% | |
| 12–17 y/o | 214,309 | 47.56% | 19,251 | 67.25% | 573 | 41.81% | |
| 18–24 y/o | 65,742 | 14.59% | 3,659 | 12.78% | 255 | 18.62% | |
| 25–34 y/o | 47,291 | 10.50% | 1,381 | 4.82% | 244 | 17.82% | |
| 35–44 y/o | 24,735 | 5.49% | 566 | 1.98% | 111 | 8.10% | |
| 45+ y/o | 11,037 | 2.45% | 239 | 0.84% | 65 | 4.78% | |
| Median Household Income | | | | | | | <0.0001 |
| 0-25th percentile | 69,853 | 15.73% | 3,210 | 11.36% | 203 | 15.13% | |
| 26th-50th percentile | 91,765 | 20.67% | 4,783 | 16.93% | 318 | 23.69% | |
| 51st-75th percentile | 119,326 | 26.88% | 7,265 | 25.71% | 291 | 21.73% | |
| 76th-100th percentile | 163,011 | 36.72% | 12,997 | 46.00% | 529 | 39.45% | |
| Primary Payer | | | | | | | <0.0001 |
| Public | 105,529 | 23.49% | 3,959 | 13.90% | 184 | 13.56% | |
| Private | 271,207 | 60.36% | 21,333 | 74.91% | 878 | 64.68% | |
| Self-pay | 51,429 | 11.45% | 1,943 | 6.82% | 195 | 14.37% | |
| Other | 21,166 | 4.71% | 1,245 | 4.37% | 100 | 7.39% | |
| Month of the year | | | | | | | <0.0001 |
| January | 20,018 | 5.08% | 1,363 | 5.25% | 65 | 5.36% | |
| February | 21,888 | 5.55% | 1,347 | 5.18% | 63 | 5.19% | |
| March | 30,912 | 7.84% | 1,948 | 7.50% | 150 | 12.39% | |
| April | 39,908 | 10.12% | 2,614 | 10.06% | 99 | 8.15% | |
| May | 41,702 | 10.57% | 2,623 | 10.10% | 123 | 10.20% | |
| June | 28,374 | 7.19% | 1,244 | 4.79% | 121 | 9.97% | |
| July | 24,024 | 6.09% | 906 | 3.49% | 84 | 6.93% | |
| August | 31,951 | 8.10% | 1,792 | 6.90% | 102 | 8.42% | |
| September | 56,801 | 14.40% | 4,367 | 16.81% | 146 | 12.09% | |
| October | 52,688 | 13.36% | 4,500 | 17.32% | 144 | 11.87% | |
| November | 28,841 | 7.31% | 2,010 | 7.74% | 89 | 7.36% | |
| December | 17,283 | 4.38% | 1,266 | 4.87% | 25 | 2.06% | |
| Day of the week | | | | | | | 0.25 |
| Weekday | 280,543 | 62.26% | 17,757 | 62.03% | 788 | 57.56% | |
| Weekend | 170,038 | 37.74% | 10,868 | 37.97% | 581 | 42.44% | |
| Hospital region | | | | | | | <0.001 |
| Northeast | 84,130 | 18.67% | 5,719 | 19.98% | 188 | 13.72% | |

*(Continued)*

**Table 1.** (Continued)

| Characteristics | All other injuries | | Concussions | | Intracranial injuries | | p-value |
|---|---|---|---|---|---|---|---|
| | n | % | n | % | n | % | |
| Midwest | 96,089 | 21.33% | 6,722 | 23.48% | 270 | 19.68% | |
| South | 105,512 | 23.42% | 7,108 | 24.83% | 443 | 32.38% | |
| West | 164,856 | 36.59% | 9,076 | 31.71% | 469 | 34.22% | |
| Hospital Teaching status ‡ | | | | | | | 0.55 |
| Non-teaching | 264,233 | 58.64% | 16,884 | 58.98% | 732 | 53.46% | |
| Teaching | 186,354 | 41.36% | 11,741 | 41.02% | 637 | 46.54% | |
| Hospital Trauma Designation | | | | | | | <0.01 |
| Not trauma | 192,797 | 42.79% | 11,376 | 39.74% | 500 | 36.52% | |
| Trauma level I, II, and III | 171,958 | 38.16% | 12,076 | 42.19% | 587 | 42.87% | |
| Non-trauma and trauma level III | 85,832 | 19.05% | 5,173 | 18.07% | 282 | 20.62% | |
| Hospital Location ‡ ‡ | | | | | | | 0.001 |
| Large Metro | 265,113 | 58.84% | 15,789 | 55.16% | 877 | 64.06% | |
| Other | 185,473 | 41.16% | 12,835 | 44.84% | 492 | 35.94% | |
| Multiple Injuries | | | | | | | <0.0001 |
| Yes | 56,509 | 12.54% | 8,615 | 30.10% | 751 | 54.82% | |
| No | 394,078 | 87.46% | 20,010 | 69.90% | 619 | 45.18% | |

‡, non-metropolitan hospitals collapsed into non-teaching, as teaching hospitals are rare in non-metropolitan areas.

‡ ‡, Not large metropolitan hospitals collapsed into other, due to scarce data in individual categories.

soccer injuries; whereas ICI are over seven-fold more likely to present to the ED with multiple injuries (Fig 1). Interestingly, similarly as multiple injuries, concussions and ICI also demonstrated notable differences in disposition.

Generally, 98% of soccer-related ED visits resulted in routine (treat-and-release) visits (Table 2). Yet, soccer-related concussion and ICI were more likely to result in hospitalization when compared to all other-soccer injuries. When compared to ED visits that ended in release to the community, concussions showed nearly 1.5-fold higher odds of transfer to a short-term hospital than did visits related to all other-soccer injuries, OR = 1.46, (1.27, 1.68). There was no difference in odds of being admitted as an inpatient to the same hospital (vs routine) between concussions and all other-soccer injuries, OR = 0.95, 95% CI [0.85, 1.07]. ICI

**Table 2. Associations between disposition of adults and children evaluated in emergency departments for soccer-related injuries using a weighted, population-based sample of visits abstracted from hospital-affiliated emergency department medical records, 2010–2013 (weighted case estimates and proportional distributions based on probability sampling).**

| | All other injuries | | Concussion | | Intracranial injuries | |
|---|---|---|---|---|---|---|
| | n | % | n | % | n | % |
| Disposition from Emergency Department | | | | | | |
| Routine (i.e., treat and release) | 441,823 | 98.08% | 28,009 | 97.85% | 883 | 64.49% |
| Transfer to short-term hospital | 2,390 | 0.53% | 221 | 0.77% | 180 | 13.13% |
| Transfer other | 400 | 0.09% | 28 | 0.10% | 17 | 1.23% |
| Home health care | 149 | 0.03% | - | - | 0 | 0.00% |
| Against medical advice | 670 | 0.15% | 54 | 0.19% | 0 | 0.00% |
| Admitted as inpatient to same hospital | 5,041 | 1.12% | 304 | 1.06% | 290 | 21.16% |
| Died in emergency department | - | - | 0 | 0.00% | 0 | 0.00% |

- = Unable to report to maintain confidentiality, ≤10 cases.

**Table 3. Associations between body injury location, mental health diagnosis, mechanism of injury, and place of injury for a population-based sample of patients presenting to U.S. emergency departments for evaluation and care for soccer related injuries (weighted case estimates and proportional distributions based on probability sampling).**

| Characteristics | All other injuries | | Concussions | | Intracranial injuries | | |
|---|---|---|---|---|---|---|---|
| | n | % | n | % | n | % | p-value |
| Mental health conditions | | | | | | | |
| ADD | 3,353 | 0.7% | 388 | 1.4% | 14 | 1.0% | <0.0001 |
| Other mental health | 10,602 | 2.4% | 866 | 3.0% | 103 | 7.5% | <0.0001 |
| Injury Location | | | | | | | |
| Head and neck | 74,183 | 16.5% | 28,625 | 100.0% | 1,369 | 100.0% | N/A |
| Trunk | 21,395 | 4.7% | 508 | 1.8% | - | - | <0.0001 |
| Upper extremities | 132,181 | 29.3% | 489 | 1.7% | 25 | 1.8% | <0.0001 |
| Lower extremities | 208,981 | 46.4% | 387 | 1.4% | 19 | 1.4% | <0.0001 |
| Mechanism of injury | | | | | | | |
| Fall | 96,960 | 21.5% | 4,660 | 16.3% | 173 | 12.6% | <0.0001 |
| Struck by hit or thrown ball, with no subsequent fall | 175,358 | 38.9% | 15,818 | 55.3% | 804 | 58.7% | <0.0001 |
| Struck by hit or thrown ball, with subsequent fall | 23,652 | 5.2% | 4,608 | 16.1% | 192 | 14.0% | <0.0001 |
| Place of injury | | | | | | | |
| Home | 7,886 | 1.8% | 206 | 0.7% | 18 | 1.3% | <0.0001 |
| Recreational/Sports facility | 216,736 | 48.1% | 14,134 | 49.4% | 719 | 52.5% | 0.19 |
| Street/Highway | 992 | 0.2% | 81 | 0.3% | 0 | 0.0% | N/A |
| Public institution (e.g. school) | 16,854 | 3.7% | 1,291 | 4.5% | 45 | 3.3% | 0.03 |
| Residential institution | 928 | 0.2% | 24 | 0.1% | - | - | 0.02 |
| Other (e.g. beach) | 15,610 | 3.5% | 893 | 3.1% | 38 | 2.8% | 0.27 |

- = Unable to report to maintain confidentiality, ≤10 cases; N/A = unable to calculate significance since some cells equal zero; Note, all variables are dichotomous (yes vs. no), which means some cases may present with multiple injuries. For example, a case may present with a head/neck injury as well as a lower extremity injury. Furthermore, we are only reporting the "yes" percentages for each variable (for example, concussions with other mental health diagnosis is 3% vs. 97% without other mental health diagnosis [which sums to 100%], but we only report the 3%).

diagnosis carried a 37-fold (95% CI [31.89, 44.39]) higher odds of transfer to a short-term hospital (vs. treat and release) compared to any other soccer injury; similarly, ICI-diagnosed patients showed over 28-fold (95% CI [25.11, 32.93]) higher odds of (same-hospital) admission to inpatient care.

Mental health conditions and mechanism of injury varied between all other-soccer injuries, concussions, and ICI are detailed in Table 3 and Fig 2. No ICI-affected patients received trunk injury diagnoses, and some affected by other soccer injuries also showed head and neck injuries other than ICI and concussion (Table 3). ICI were 1.33-fold to be identified with ADD and over two-fold more likely to be identified with other mental health conditions when compared to all other injuries. Concussions were 1.38-fold to be identified with ADD and 1.55-fold more likely to be identified with other mental health conditions when compared to all other injuries. When looking at mechanism of injury, soccer players were nearly two-fold more likely to be diagnosed with a concussion following a fall; over 3-fold when struck by hit or ball without a subsequent fall; and nearly 7-fold when struck by hit or ball with a subsequent fall. Furthermore, soccer players were nearly 1.4-fold more likely to be diagnosed with an ICI following a fall; over 3-fold when struck by hit or ball without a subsequent fall; and over 5-fold when struck by hit or ball with a subsequent fall. As expected, those diagnosed with upper extremity and lower extremity were less likely to be diagnosed with concussion or ICI.

| | Concussions | | | | | Other intracranial injuries | | | | |
| | 95% CI | | | Less likely | More likely | | 95% CI | | | Less likely | More likely |
| Characteristics | OR | Low | High | | | OR | Low | High | | |
|---|---|---|---|---|---|---|---|---|---|---|
| Mental health conditions | | | | | | | | | | |
| Attention deficit disorder (ref: no) | 1.38 | 1.07 | 1.77 | | | 1.33 | 0.40 | 4.46 | | |
| Other mental health (ref: no) | 1.55 | 1.32 | 1.83 | | | 2.44 | 1.45 | 4.10 | | |
| Injury Location | | | | | | | | | | |
| Upper extremities (ref: no) | 0.016 | 0.013 | 0.019 | | | 0.018 | 0.008 | 0.05 | | |
| Lower extremities (ref: no) | 0.008 | 0.006 | 0.010 | | | 0.008 | 0.003 | 0.03 | | |
| Mechanism of injury | | | | | | | | | | |
| Fall (ref: no) | 1.97 | 1.75 | 2.22 | | | 1.41 | 0.83 | 2.38 | | |
| Struck by hit or thrown ball, no subsequent fall (ref: no) | 3.42 | 3.03 | 3.85 | | | 3.12 | 2.09 | 4.65 | | |
| Struck by hit or thrown ball, subsequent fall (ref: no) | 6.84 | 5.93 | 7.90 | | | 5.12 | 3.03 | 8.65 | | |
| Place of injury | | | | | | | | | | |
| Home (ref: no) | 0.57 | 0.41 | 0.79 | | | 1.12 | 0.33 | 3.74 | | |
| Recreational/Sports facility (ref: no) | 1.18 | 1.09 | 1.27 | | | 1.20 | 0.84 | 1.70 | | |
| Public institution (e.g. school) (ref: no) | 1.37 | 1.16 | 1.62 | | | 1.38 | 0.67 | 2.85 | | |
| Other (e.g. beach) (ref: no) | 1.07 | 0.90 | 1.26 | | | 0.84 | 0.36 | 1.91 | | |

*Simultaneously adjusting for the effects of sociodemographic, geographic and health care resources. Weighted case estimates and proportional distributions based on probability sampling.

**Fig 2. Adjusted odds of concussion or intracranial injuries (each) versus all other soccer injuries evaluating the effect of exposure to two mental health conditions, two injury locations, three mechanism of injury, and four place of injury (each) in a population-based sample of persons receiving evaluation and care at U.S. emergency departments for soccer related injuries.**

Adjusted logistic regression models demonstrated notable differences in odds of presenting to the ED with a soccer-related concussion and ICI compared to all other-soccer injuries by year, gender, and age. For example, soccer concussions (vs. all-other soccer injuries) adjusted odds ratios, with 2010 as the reference year, increased yearly (Fig 1), from 1.13-fold to 1.30-fold. ICI (vs all other-soccer injuries) odds ratio decreased between 1.20-fold to 1.85-fold annually over the observation period. Females had significantly higher odds of presenting with a concussion than all-other soccer injury, OR = 1.16 (95% CI [1.09, 1.24]), but nearly two-fold lower odds of presenting with an ICI, OR = 0.56 (95% CI [0.40, 0.79]). As for age, 12–17 y/o had higher odds of presenting with a concussion across all age groups; for example, they were 1.49-fold and over 4-fold higher odds when compared to 18–24 or 45+ y/o patients with soccer-related injury. The odds of ICI were not significantly different when 12–17 y/o were compared to most other age groups; however, youth 7–11 y/o showed lower odds of ICI than12-17 y/o (OR = 0.54, 95% CI [0.33, 0.87]).

With the exception of ICI, higher socioeconomic status in the form of income and insurance coverage may influence whether or not patients present to ED following a soccer-related concussion. Generally, ED visits for highest income earners (76th-100th percentile) had higher odds of presenting to the ED following a concussion compared to those reporting ≤25th, 26th-50th, and 51st-75th percentiles, with odds of 1.45-fold, 1.42-fold, and 1.22-fold, respectively. Privately-insured individuals showed nearly two-fold higher odds of presenting to the ED with a soccer-related concussion, vs all-other injuries, than those with public insurance; and 1.37-fold higher compared to self-pay. There was no significant difference in odds of presenting to the ED with a soccer-related concussion (vs all-other injuries) between privately insured individuals and other forms of payer. Noteworthy, there was no significant differential odds of presenting with an ICI compared to all-other injuries between the highest household income quartile or other forms of insurance also did not show significant odds differences of presenting with an ICI to the ED when compared to all-other injuries, with the exception of public insurance which showed 1.72-fold lower odds.

Odds of concussions compared to all-other soccer injuries was lower in June and July, 1.41-fold and 1.56-fold, respectively, when compared to January. Generally, the differences in

odds of presenting to the ED with a concussion or ICI did not vary across weekdays and week-ends, and relationships between regions, types of hospitals, trauma-designated centers, and large metropolitan population densities were small relative to their comparators (Fig 1).

ED visits with concussions or ICI were associated with multiple injuries. When compared to all-other soccer injuries, those with concussion or ICI were nearly three- and seven-fold more likely to present with multiple injuries (i.e., OR = 2.92, 95% CI [2.70, 3.16]; OR = 7.55, 95% CI [5.73, 9.94], respectively).

Patients spent more than $700 million (U.S.) across four years, 2010–2013, for soccer-related injuries presenting to ED's. Generally, total charges per person for ICI visits were more than 7-fold higher than those for all-other injuries. When compared to cost for concussion, ICI costs were over 5-fold higher (Table 4). Total charges, per person, were 40% higher for concussion-affected youth and adults than for all-other soccer injuries. For total inpatient care of soccer-related injuries, ICI charges were more than 2-fold higher than for similarly hospitalized patients with concussion. Inpatient length of stay was 34% longer for ICI-affected patients injured in soccer play than all-other soccer injuries and nearly 3-fold longer than those suffering concussions (Fig 3).

## Discussion

This study described (1) soccer-related concussions, ICI, and all-other soccer injuries leading to U.S. ED visits, as well as (2) healthcare utilization, and (3) analyzed independent predictors of concussions, ICI, and all-other soccer injuries leading to ED visits. To our knowledge, this is the first study that provides U.S. national estimates of ED visits, using a stratified and clustered probability weighted sample of hospital-owned ED's, for soccer-related concussions, ICI, and all-other soccer injuries. The findings of the study are important for policy makers, clinicians, coaches and families involved in soccer communities, as well as future research that may help reduce soccer-related injuries and ED utilization.

The data demonstrated a steady increase of concussions and all-other soccer injuries lead-ing to U.S. ED visits over the four-year period; interestingly, ICI showed a steady decline over time. Other data have shown similar results; for example, trends from 1990 to 2003 in non-stratified analysis determined an overall increase in soccer-related incidence rate (IR), 1.6/1,000 of ED visits vs 1.7/1,000 (p = 0.78), respectively [15]. Other have shown that soccer-related TBI injury rates nearly doubled between 2001 and 2012, and account for 2.9% all of soccer-related ED visits in 5–9 y/o, followed by 1.5% in 10–14 y/o, and reaches 3.7% peak in 15–19 y/o [16, 17]. In a more recent analysis, soccer-related injury rates increased over 100% in a 25-year period, whereas concussion/closed head injuries increased by 1595% over the

**Table 4. Charges and length of stay for a population-based sample of patients presenting to U.S. emergency departments for evaluation and care for soccer related injuries (weighted case estimates, and proportional distributions based on probability sampling).**

| Characteristics | All other Injuries | | Concussions | | Intracranial injuries | |
|---|---|---|---|---|---|---|
| | $ | SD | $ | SD | $ | SD |
| Total charges per person, in thousands | 1,434.24 | | 2,017.89 | | 10,613.59 | |
| ED charges, in million | | | | | | |
| Total charges | 479.43 | (23.03) | 51.66 | (2.70) | 0.34 | (0.34) |
| Mean | 1,515 | (33.71) | 2,379 | (89.04) | 3,300 | (234.30) |
| Inpatient charges | | | | | | |
| Total charges, in million | 166.82 | (10.55) | 6.10 | (1.08) | 14.19 | (4.27) |
| Mean | 34,317 | (1,301) | 20,303 | (2,120) | 49,822 | (13,582) |
| Inpatient length of stay | | | | | | |
| Mean, days (SD) | 2.55 | (0.09) | 1.23 | (0.13) | 3.43 | (0.78) |

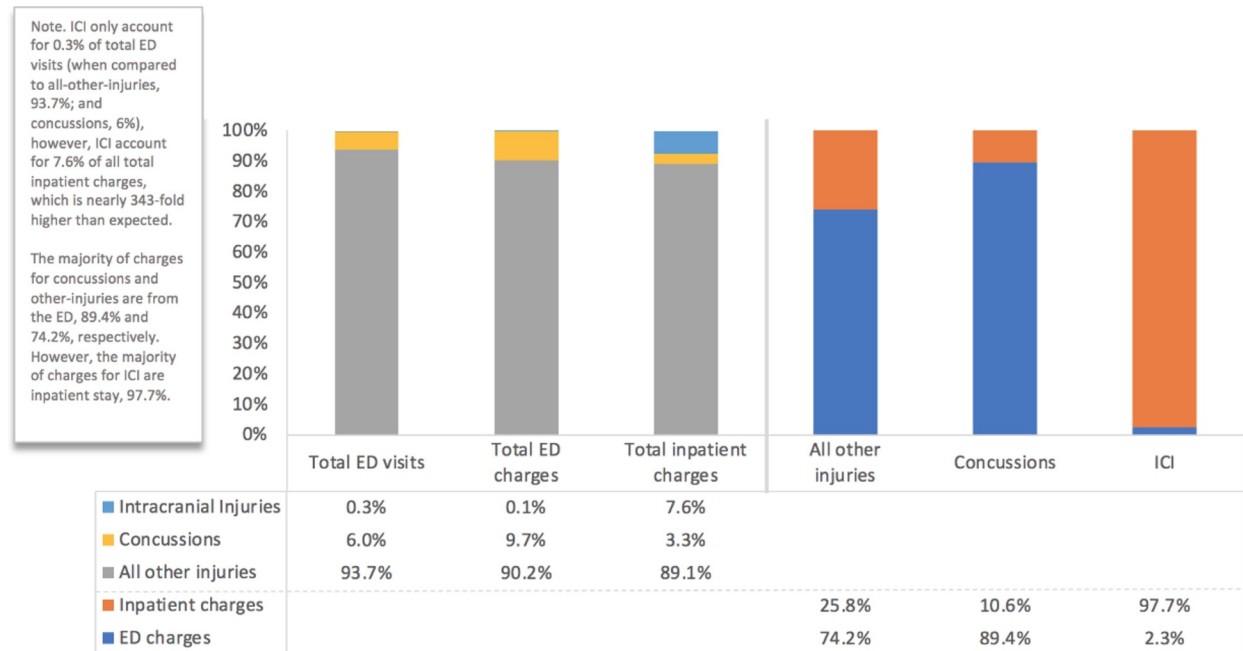

**Fig 3. Total Emergency Department (ED) visits, ED charges, and inpatient charges for a population-based sample of patients presenting to U.S. emergency departments for evaluation and care for soccer related concussions, intracranial injuries (ICI) and all-other injuries, as well as proportion of inpatient vs. ED charges within those diagnoses (national weighted case estimates).**

same period [18]. The increased ED rates of soccer-related concussion may be attributed to a combination of increase in soccer participation, greater awareness and detection of concussions in soccer, and broader definition of concussion in sport [1, 3, 4, 19–21]. In contrast to other findings, the NEDS data set demonstrated steady decline of ICI over time. It was also noted that concussions make up a larger percentage of female injuries than males; these findings can be interpreted in context of higher rates of concussion in females. Incidence studies of sports concussions per 'athlete exposures' (games or practices) have demonstrated that for sports with similar rules for both genders, such as soccer, the rates of concussion are higher in females than males [22–25]. Further research is needed to better understand the reason for the decline of soccer-related ICI as a diagnostic outcome over time, as well as individual differences.

Generally, diagnoses of soccer-related concussions and ICI are uncommon in ED visits, in comparison to all-other injuries. Soccer-related concussion rates reported by others vary from 1.38 to 3.10/10,000 participants, and account for less than 4% of ED visits [3, 10, 16]. However, in the estimated 3.4 million sports- and recreation-related TBIs, among 15–19 y/o males and females, soccer-related TBI's were proportionally in the top four sport-related activities leading to TBI ED visits [17]. In this study, concussions leading to ED visits accounted for nearly 6% of all soccer-injuries, whereas ICI accounted for 0.28%. Nevertheless, generally, the results demonstrate that most soccer-related injuries that lead to ED visits can be treated outside of the ED, as the majority are treat and release.

Soccer-related hospitalization is rare for concussions and all other injuries, but not for ICI. The data showed that less than 2% of cases presenting to ED's after a soccer-related injury are hospitalized; surprisingly nearly 35% of soccer-related ICI ED visits result in hospitalization. To our knowledge, there is no study that has described ICI proportion of hospitalization, most focus on disposition of general injuries. Others have similar results, with less than 2% ED visits

resulting in hospitalization [10, 15, 16, 26]. This evidence suggests that the public may inadvertently misuse U.S. ED for non-emergent injuries that can be treated in other settings, such as outpatient or in-field, which may be due to public misunderstanding of ED use or lack of access to appropriate treatment centers. There may be some factors that future studies may consider and incorporate, which are not available in this administrative data. For example, weekday injuries may have more access to their primary care providers clinic, which means player will not go ED, and weekend injuries may not be able to see primary care provider, which may more likely result in ED visit. Moreover, the literature does not report the prevalence or rate of soccer-related hospitalizations due to brain trauma, which may account for a large number of needed hospitalizations, as well as high cost to the public.

Soccer-related injuries are a financial burden to the public. The unique contribution of this work is the inclusion of total charges and length of stay of soccer-related concussions, ICI, and all-other injuries leading to U.S. ED visits; To our knowledge, no other study has looked at charges and length of stay in soccer-related injuries. ICI inpatient length of stay was nearly 3-fold compared to concussions, and 34% higher than all-other injuries. As it relates to financial cost, soccer-related injuries charges average approximately $180 million annually in ED visits (and hospitalization resulting from the ED visit). The cost ranged depending on the reason for the visit, with ICI total charges per person more than 5-fold higher than concussions, and 7-fold higher than all other soccer injuries. Interestingly, nearly all the charges for ICI resulted from inpatient charges (97.7%), whereas the majority of charges for concussions (89.4%) and other-injuries (74.2%) were from ED charges. It should be noted that the soccer-related injury charges are limited to U.S. ED visits (and any additional care and services resulting from the ED visit), cost reported in this study do not include any charges that may be incurred in other forms of health care and treatment for injuries, such as outpatient clinics, over the counter medications and treatment, and rehabilitation (e.g. physical therapy), among others. The cost of soccer-related injuries is higher than reported, and this analysis only focuses on ED visits in an effort to provide information to policy-makers and clinicians that specialize in ED settings.

Adjusted logistic regression models showed that indicators of higher socio-economic status, median household income and private insurance, independently predicted visiting an ED following a soccer-related concussion, which is often seen as a serious injury with sequelae, as compared to all-other injuries. Interestingly, those with ICI tended to visit ED regardless of socioeconomic indicators. For example, while tabular analyses suggest that high-income and privately-insurance increases the likeliness of ED admission for soccer-related injuries, we found the no statistically significant effect of income on risk for ICI. However, all other income groups were statistically significantly less likely to be diagnosed with concussion than those from high income households. However, the source of primary payment for service did affect some relationships. For instance, only publicly insured patients were statistically significantly less likely those privately insured to be diagnosed with ICI. However, both publicly funded and self-pay patients were less often diagnosed with concussion than those privately insured. This may be due to the seriousness of injuries with ICI; Hence, people may be more willing to attend the ED visit regardless of cost given the potential seriousness sequelae.

Concussions and ICI may present with multiple injuries when compared to all-other injuries. However, the difference when compared to other-soccer-injuries is greater for ICI than concussions. Concussions were nearly 3-fold more likely to present to the ED with multiple injuries when compared to all-other injuries; whereas, ICI were more than 7-fold more likely to present with multiple injuries. Clinicians should assess for multiple injuries when a patient presents with a head injury such as a concussion and ICI.

## Limitations

The unique contribution of this work is the inclusion of descriptors, disposition, total charges, and length of stay of soccer-related concussions, ICI, and all-other injuries leading to U.S. ED visits. These methods of analysis allowed greater generalizability and uncover preventable injury patterns susceptible to targeted public health interventions. Nonetheless, limitations reflect that ED data may under-estimate the prevalence and rate of certain soccer-related injuries experienced in the field, as affected individuals may not present to the ED [27], and does not distinguish injuries related to different forms of soccer play, field-soccer vs futsal vs indoor. Unfortunately, the NEDS dataset does not include differences in forms of soccer-play, or exposure measures, e.g. head and lower extremity impacts. NEDS data cannot determine whether some groups are more or less likely to be injured, seek ED care (if injured), or receive similar care once admitted. For example, if household income and payer source increase the likelihood of some diagnostic procedures, the number of concussion or ICI diagnoses may increase. Alternatively, income may be associated with risk taking among some youth and adults. As a result, analyses were limited to the administrative data available in NEDS, and differences in forms of soccer play were not considered.

The cross-sectional design of administrative data limits causation directionality between variables; in addition, incidence rate cannot be determined. It is important to note that results should be taken with caution, and future research should think of ways to determine if the magnitude of the ED numbers may be driven by participation numbers, including the gender and age effects. To better understand the epidemiology of U.S. ED soccer-related injuries hospitalization, cost, and length of stay, future studies should consider analyzing difference of field-injury exposure in different forms of soccer play (e.g. field-soccer, futsal, indoor), as well as prospective data collection that considers soccer participation compared to injury.

In conclusion, given that soccer-related injuries have become a health and healthcare utilization and cost concern, the current study allowed insight into soccer related concussion, ICI, and all-other injury descriptors, hospitalization, ED and inpatient charges, and hospital length of stay, across individual-level, hospital-level, and multiple injuries in soccer-related injuries. The finding helped determine multi-level features susceptible to targeted injury prevention intervention to minimize the price on human suffering, healthcare cost, and utilization, within concussions, ICI, all-other soccer injuries leading to an ED visit.

## Author Contributions

**Conceptualization:** Gerardo Flores, Christopher C. Giza, Barbara Bates-Jensen, Mary-Lynn Brecht, Dorothy Wiley.

**Formal analysis:** Gerardo Flores, Christopher C. Giza, Barbara Bates-Jensen, Mary-Lynn Brecht, Dorothy Wiley.

**Investigation:** Gerardo Flores.

**Methodology:** Gerardo Flores, Dorothy Wiley.

**Resources:** Dorothy Wiley.

**Software:** Gerardo Flores, Dorothy Wiley.

**Supervision:** Gerardo Flores, Dorothy Wiley.

**Writing – original draft:** Gerardo Flores, Christopher C. Giza, Barbara Bates-Jensen, Mary-Lynn Brecht, Dorothy Wiley.

**Writing – review & editing:** Christopher C. Giza, Barbara Bates-Jensen, Mary-Lynn Brecht, Dorothy Wiley.

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
