## [Decision Letter · Decision Letter 0]

13 Apr 2021

PONE-D-21-06957

Soccer-related injuries utilization of U.S. emergency departments for concussions, intracranial injuries, and other-injuries in a national representative probability sample: Nationwide Emergency Department Sample, 2010 to 2013

PLOS ONE

Dear Dr. Flores,

Thank you for submitting your manuscript to PLOS ONE. After careful consideration, we feel that it has merit but does not fully meet PLOS ONE’s publication criteria as it currently stands. Therefore, we invite you to submit a revised version of the manuscript that addresses the points raised during the review process.

It is an interesting and nicely done investigation. By th way, ths study may help to make soccer even morepopulat in the United States. The comments made by reviewer 2 are important. The authors shoulb follow his recommendations to improve the manuscript: The medical descriptions, diagnostic outcomes as well as the mechanisms of these injuries shoulb be explained more in details.

We look forward to receiving your revised manuscript.

Kind regards,

Hans-Peter Simmen, M.D., Professor of Surgery

Academic Editor

PLOS ONE

Journal Requirements:

PLOS requires an ORCID iD for the corresponding author in Editorial Manager on papers submitted after December 6th, 2016. Please ensure that you have an ORCID iD and that it is validated in Editorial Manager. To do this, go to ‘Update my Information’ (in the upper left-hand corner of the main menu), and click on the Fetch/Validate link next to the ORCID field. This will take you to the ORCID site and allow you to create a new iD or authenticate a pre-existing iD in Editorial Manager. Please see the following video for instructions on linking an ORCID iD to your Editorial Manager account: https://www.youtube.com/watch?v=_xcclfuvtxQ

In ethics statement in the manuscript and in the online submission form, please provide additional information about the patient records/samples used in your retrospective study. Specifically, please ensure that you have discussed whether all data/samples were fully anonymized before you accessed them and/or whether the IRB or ethics committee waived the requirement for informed consent. If patients provided informed written consent to have data/samples from their medical records used in research, please include this information.

Please ensure that you refer to Figure 2 in your text as, if accepted, production will need this reference to link the reader to the figure.

Reviewers' comments:

Reviewer's Responses to Questions

**Comments to the Author**

1. Is the manuscript technically sound, and do the data support the conclusions?

Reviewer #1: Yes

Reviewer #2: Partly

2. Has the statistical analysis been performed appropriately and rigorously? 

Reviewer #1: Yes

Reviewer #2: Yes

3. Have the authors made all data underlying the findings in their manuscript fully available?

Reviewer #1: Yes

Reviewer #2: Yes

4. Is the manuscript presented in an intelligible fashion and written in standard English?

Reviewer #1: Yes

Reviewer #2: Yes

5. Review Comments to the Author

Reviewer #1: You describe a very intresting overview of more than 480'000 of US Emergency Departments visits in the years 2010 to 2013. For the future it will be important to see, if the soccer-related injuries will change. In this kind it would be of interest to have actual statistics, not only 8 and more years ago, also of othopedic injuries. But I think, that we will not have important changes in the main statements. So you give important recommendations to a lot of people, as you describe to policy makers, economists, clinicians, coaches and families involved in soccer communities. The target will be to reduce injuries, especially serious accidents as for example concussions and intracranial brain injuries, which will often have a consequence of definitive sports disability in repeated cases. Special attention needs the age group of 12-17 y/o (males and females) with much more risk. A very important statement not only to the coaches but also to FIFA, continental and national federations, which organize youth championships and tournaments. The costs of soccer-related injuries will be much higher than reported, and for future reports it will be interesting, not only for policy makers and economists, but also for coaches, physiotherapists, players etc, to know the importance of accompanying measures in soccer injuries. You will have a lot of work in the future relating to soccer.

Reviewer #2: The authors have done a great analysis first of all on the description of the healthcare utilization, cost and length of hospital stay. Compliment. The independent predictors of these injuries have also been described in well done way. Two of the primary purposes of this article have been reached. Table 1 and 2 are well done.

However, the third targets of the manuscript, such as the medical descriptions, the diagnostic outcomes and the mechanisms of these injuries has not been investigated enough in details. The table 3 should also been ameliorated and it should be explained more in details. The authors have to describe and discuss more meticulous the relationship between all injuries/concussion/ICI and the injury localization on one hand and the mechanism of the injuries on the other hand.

6. PLOS authors have the option to publish the peer review history of their article (what does this mean?). If published, this will include your full peer review and any attached files.

Reviewer #1: No

Reviewer #2: No

---

## [Author Response · Author response to Decision Letter 0]

2 Sep 2021

Thank you for the useful feedback. based on the feedback, we performed additional adjusted logistic regressions that provided greater depth to the study (please see new figure 2)

---

## [Editor Report · Decision Letter 1]

27 Sep 2021

Soccer-related injuries utilization of U.S. emergency departments for concussions, intracranial injuries, and other-injuries in a national representative probability sample: Nationwide Emergency Department Sample, 2010 to 2013

PONE-D-21-06957R1

Dear Dr. Flores,

We’re pleased to inform you that your manuscript has been judged scientifically suitable for publication and will be formally accepted for publication once it meets all outstanding technical requirements.

Kind regards,

Hans-Peter Simmen, M.D., Professor of Surgery

Academic Editor

PLOS ONE
---

## [Editor Report · Acceptance letter]

4 Oct 2021

PONE-D-21-06957R1 

Soccer-related injuries utilization of U.S. emergency departments for concussions, intracranial injuries, and other-injuries in a national representative probability sample: Nationwide Emergency Department Sample, 2010 to 2013 

Dear Dr. Flores:

I'm pleased to inform you that your manuscript has been deemed suitable for publication in PLOS ONE. Congratulations! Your manuscript is now with our production department. 

Kind regards, 

on behalf of

Dr. Hans-Peter Simmen 

Academic Editor

PLOS ONE